# Breakage Strength of Wood Sawdust Pellets: Measurements and Modelling

**DOI:** 10.3390/ma14123273

**Published:** 2021-06-13

**Authors:** Józef Horabik, Maciej Bańda, Grzegorz Józefaciuk, Agnieszka Adamczuk, Cezary Polakowski, Mateusz Stasiak, Piotr Parafiniuk, Joanna Wiącek, Rafał Kobyłka, Marek Molenda

**Affiliations:** Institute of Agrophysics, Polish Academy of Sciences, Doświadczalna 4, 20-290 Lublin, Poland; m.banda@ipan.lublin.pl (M.B.); g.jozefaci@ipan.lublin.pl (G.J.); a.adamczuk@ipan.lublin.pl (A.A.); c.polakowski@ipan.lublin.pl (C.P.); m.stasiak@ipan.lublin.pl (M.S.); p.parafiniuk@ipan.lublin.pl (P.P.); j.wiacek@ipan.lublin.pl (J.W.); r.kobylka@ipan.lublin.pl (R.K.); m.molenda@ipan.lublin.pl (M.M.)

**Keywords:** sawdust pellets, diametral compression test, tensile strength, bonded particle model

## Abstract

Wood pellets are an important source of renewable energy. Their mechanical strength is a crucial property. In this study, the tensile strength of pellets made from oak, pine, and birch sawdust with moisture contents of 8% and 20% compacted at 60 and 120 MPa was determined in a diametral compression test. The highest tensile strength was noted for oak and the lowest for birch pellets. For all materials, the tensile strength was the highest for a moisture content of 8% and 120 MPa. All pellets exhibited a ductile breakage mode characterised by a smooth and round stress–deformation relationship without any sudden drops. Discrete element method (DEM) simulations were performed to check for the possibility of numerical reproduction of pelletisation of the sawdust and then of the pellet deformation in the diametral compression test. The pellet breakage process was successfully simulated using the DEM implemented with the bonded particle model. The simulations reproduced the results of laboratory testing well and provided deeper insight into particle–particle bonding mechanisms. Cracks were initiated close to the centre of the pellet and, as the deformation progressed, they further developed in the direction of loading.

## 1. Introduction

Waste biomaterials, including wood processing residues, are important renewable energy resources [1,2]. Since, in their native loose forms, such materials have low bulk density and their storage and transport are difficult and costly [3], densification is applied to reduce these disadvantages [4]. Biomaterials can be densified into briquettes, pellets, or cubes, among which pelletising is the most popular. The quality of the pellets is related to their physical, chemical, and mechanical properties. Under high-pressure pelletisation, the bulk density of wood sawdust increases from 100–200 to 1000–1200 kg m^−^^3^, which is the typical density of the pellets [5,6]. The bulk density of pellets is significantly higher than that of natural wood block (400–700 kg m^−^^3^); however, it is lower than the solid-phase density of sawdust particles (1450 kg m^−^^3^) [7]. Hydrogen binding at the surfaces of lignocellulosic particles of sawdust provides the main type of binding force used in manufacturing of pellets [1]. With increasing compaction pressure, the contact area between the particles increases and the pellets become denser and more durable. Binding forces are higher when a contact zone between particles covers a larger area; that is, the forces increase with an increase in compaction pressure and a decrease in particle size [8]. To increase binding forces, steam explosion pretreatment is applied; this causes lignin melting and a reduction in particle size [3]. The moisture content (*MC*) markedly affects the binding forces. It has both antagonistic (with water molecules replacing wood polymer bonds) and protagonistic (decreasing the melting temperature of lignin) actions on pellet durability [2]; therefore, an optimum *MC* should be maintained during the pelletisation process [9]. Pressure (contact between particles), temperature (plastic deformation of lignin), particle size, and optimum *MC* are key factors in particle binding [1,3]. Therefore, properly pretreated materials have better physical quality than untreated materials [2,6].

Mechanical durability is a crucial property of pellets [10,11]. However, very simple and convenient standard durability tests provide only a qualitative measure of the mechanical strength of the pellets [12]; therefore, diametral compression can be alternatively applied. This method is also very simple [13] and has been widely used to determine the strength of cylindrical samples by measuring the direct relationship between the load at failure and tensile strength [14]. This test has been used to determine the tensile strength of rocks, concrete, composites, and agglomerates in numerous branches of science and technology, such as civil and chemical engineering [15,16], pharmaceutical science [17,18,19,20], biomass briquettes and pellets engineering [4,6,11,21,22], and food powder engineering [23,24]. Larsson and Samuelsson [11] used diametral compression to show that the tensile strength of pellets was strongly correlated with their density and durability. Using the diametral compression test, Shaw et al. [6] demonstrated that steam treatment leads to an increase of more than tenfold in the tensile strength of poplar wood pellets.

The discrete element method (DEM) proposed by Cundall and Strack [25] has led to new possibilities for modelling interactions and processes between a large number of individual objects. The method has been applied in numerous areas of science and technology, including physics [26], pharmaceutical science [27], civil engineering [28], and agricultural engineering [29]. This method is a very promising tool for modelling biomass post-harvest processing, including handling [30,31,32], compaction [33], and testing biomass bulk properties [34]. The DEM extended with the bonded particle model (BPM) considerably broadened the range of applicability of the DEM to model the internal structure and breakage of agglomerates composed of numerous particles [35,36]. The method has been applied to model the bulk behaviour of deformable biomass fibres [34], packing of flexible fibres [37], compaction of wood chips [38], durability [39], and the strength of pellets [40]. Xia et al. [41] provided a comprehensive discussion of the challenges of modelling the behaviour of the bulk of complex-shaped particles of wood biomass by using the DEM. It seems that DEM modelling of biomass agglomerate breakage is still in an early stage of development, and additional work is needed to demonstrate the validity and limitations of the method.

The aim of this study was to investigate the possibility of numerical reproduction of the pelletisation process of sawdust and the stress–deformation behaviour of pellets in diametral compression using the DEM with a parallel BPM.

## 2. Experiments

### 2.1. Materials and Methods

#### 2.1.1. Materials

Sawdust from birch (*Betula* L.), oak (*Quercus* L.), and pine (*Pinus sylvestris* L.) wood obtained from a local sawmill were used for the experiments. The mean particle sizes determined using a set of 0.1 to 2 mm sieves were 0.355 ± 0.028, 0.409 ± 0.024, and 0.375 ± 0.026 mm for birch, oak, and pine sawdust, respectively. The initial *MC* of all sawdust types was 8% ± 0.7%. To increase the *MC* to 20 ± 0.9%, sawdust samples were placed in a chamber humidifier at a relative humidity of 70% and a temperature of 21 °C and occasionally shaken for 4 days.

#### 2.1.2. Pellets Preparation

A high-pressure piston-and-mould compaction process was used to densify the sawdust. Samples of sawdust (0.5 g) were compacted in a steel cylindrical mould with a diameter of 10 mm and height of 25 mm up to a pressure of 60 or 120 MPa using a material testing machine (Instron 7782, High Wycombe, UK) with a piston moved at 0.033 mm s^−1^. To remove the pellet, the base of the mould was removed and the pellet was pressed out [42]. Twelve combinations of pellets were prepared utilising three types of wood, two levels of moisture content, and two levels of compaction pressure. Ten specimens of each variant of pellets were prepared.

#### 2.1.3. Determination of Solid-Phase and Bulk Densities

The solid-phase density of the sawdust particles, *ρ**_s_*, and of the intact wood (40 mm diameter and 70 mm height blocks), *ρ**_w_**,* was measured using helium pycnometry (Ultrapyc 1200e, Quantachrome Instruments, Boynton Beach, FL, USA) in five replicates.

The bulk density of the intact wood, *BD_w_*, of the pellets prepared at both levels of compaction pressure, *BD_p_*, and the initial bulk density of the sawdust (just before densification), *BD_s_*, were determined from the mass and volume of the specimens. The volume of the pellets was determined from their diameter and height measured just before the diametral compression test using an electronic calliper with an accuracy of 0.01 mm. The volume of the sawdust was determined from the mould diameter and initial height of the bulk of the sawdust placed in the mould.

#### 2.1.4. Estimation of Porosity and Pore Size Distribution

The total porosity *p_x_* of the studied materials was estimated from their bulk density *BD_x_* as
(1)$$p_{x}=1-\frac{BD_{x}}{\rho_{s}}$$
where *ρ**_s_* is the solid density of the sawdust, the index *x* equals *w* for the intact wood and *p* for pellets, and *s* is the initial bulk of the sawdust. In all cases, the density of the solid phase was assumed to be equal to the density of the sawdust.

Mercury intrusion porosimetry tests were performed to estimate the structural features of the wood and pellets. Prior to the measurements, the specimens were dried overnight at 105 °C. The tests were performed for pressures ranging from ~0.1 to 200 MPa (corresponding to pore radii from ~10.0 µm to 3.8 nm) using an Autopore IV 9500 porosimeter (Micromeritics, Norcross, GA, USA) in two replicates. The intrusion volumes were measured at stepwise increasing pressures, allowing equilibration at each pressure step. The maximum deviations between the replicates were ≤4.5%, and they occurred mainly at low pressures (i.e., for the largest pores). The volume of mercury, *V* (in m^3^ kg^−1^), intruded at a given pressure *P* (in pascals) gave the pore volume that could be accessed. The intrusion pressure was translated on the equivalent pore radius *R* (in metres) using the Washburn equation
(2)$$P=\frac{-A\sigma_{m}\cos\alpha_{m}}{R},$$
where *σ**_m_* is the mercury surface tension, *α**_m_* is the mercury–solid contact angle (taken as 141.3° for all studied materials), and *A* is a shape factor (equal to 2 for the assumed capillary pores).

By knowing the dependence of *V* on *R*, a normalised pore size distribution, *f*(*R*), was calculated and expressed on a logarithmic scale [43] as:(3)$$f(R)=\frac{1}{V_{\max}}\frac{dV}{{}_{}d\log(R)}.$$

#### 2.1.5. Scanning Electron Microscopy

Scanning electron microscopy (SEM) images of the surfaces of wood blocks, sawdust, and broken pellets were taken in five replicates using a Thermo Scientific Phenom ProX G6 microscope (Thermo Fisher Scientific, Waltham, MA, USA) operating at 10 kV, and the results were analysed using the software provided by the manufacturer. Prior to SEM analysis, all samples were sputtered with gold using a CCU-010 system (Safematic, Zizers, Switzerland).

#### 2.1.6. Diametral Compression Test

The tensile strength of the pellets was determined by diametral compression between two parallel plates using a material testing machine (Lloyd LRX, Advanced Test Equipment Corp., San Diego, CA, USA) in five replicates. The displacement velocity of the plate was 0.033 mm s^−^^1^. The load was measured with an accuracy of ±0.02 N.

The stress components in the pellet during the diametral compression test are shown in Figure 1. In the centre of the pellets (where the geometrical coordinates are *x* = 0 and *y* = 0), the stresses *σ**_x_* and *σ**_y_* are the principal stresses *σ*_1_ and *σ*_2_, respectively [14]. The tensile strength at failure, *σ**_f_*, is identified by the maximum tensile stress *σ*_1,max_ in the direction perpendicular to the load in the centre of the object [13]:(4)$$\sigma_{f}=\sigma_{1,\max}^{}=\frac{P_{f}}{\pi R_{p}h_{p}},$$
where *P_f_* is the failure load, *R_p_* is the radius of the pellet, and *h_p_* is its height.

#### 2.1.7. Statistics

The data were statistically analysed using Statistica v. 10.0 (StatSoft Inc., Kraków, Poland) software and presented as mean values plus or minus the standard deviation (SD). One-way analysis of variance along with Tukey’s test with a significance level of 5% was applied to determine the statistical significance of the differences between the mean values.

### 2.2. Experimental Results

The densities of wood and sawdust are listed in Table 1. The high solid-phase density of the sawdust was probably the closest to the true solid-phase density of the wood itself. We surmised that the closest packing of the main wood component, cellulose, was similar to that of crystalline sugar (e.g., saccharose) composed of the same chemical subunits. The density of the solid phase of saccharose (1.59 g cm^−3^ [44]) is close to the densities of all the types of sawdust studied, whereas the density of the wood with its closed pores inside is markedly lower. This may indicate that the vast majority of closed wood pores were damaged during sawdust preparation, with only a small portion eventually remaining. Therefore, the density of the sawdust particles was considered as the true density of the wood solid phase. This was the reason for applying its value for bulk density calculations in Equation (1). The lower solid-phase density of the intact wood indicates that its structure contained closed pores that were not available for gas penetration. The volumetric percentages of these pores, calculated using data from Table 1, were 26.8 ± 0.3 for oak, 2.9 ± 0.2 for pine and 6.5 ± 0.1 for birch.

The bulk density and porosity of the studied intact wood, sawdust, and pellets are presented in Table 2. The porosity of the intact wood ranged from 53.8% (oak) to 65.2% (pine). The initial porosity of the bulk sawdust was ~80% for all the samples. Compaction of the sawdust resulted in a profound reduction of porosity: that of the pelletised samples ranged from 58.4% for pine (*MC* = 20%, with 60 MPa compaction) to 31.1% for oak (*MC* = 8%, with 120 MPa compaction). The porosity of the pellets was considerably lower than that of intact wood. The reduction in the porosity of the sawdust during pelletisation can be considered as a combination of two factors: continuous reduction of external pores between particles and closing of the internal pores present inside sawdust particles during the final stages of compaction [38]. At higher *MC*, the bulk density of the sawdust and the pellets was lower, which was most probably caused by the swelling of wood cells and/or by hydration of cellulose molecules. As a direct consequence of the decrease in the bulk density, the calculated porosity at higher moisture levels was higher. However, the porosity value may be uncertain because the density of the solid phase (sawdust) measured at 8% moisture was used. The water-involving processes described above may also alter the solid-phase density. Other researchers have reported porosities of bulk sawdust, intact wood, and highly compacted pellets of ~85%, ~45–60%, and ~15%–30%, respectively [5,6,7,11], which are consistent with the results of the present study.

The dependencies between the pore volume and pore radius and the pore size distribution functions for the studied wood and pellets are presented in Figure 2 and Figure 3, respectively.

From both figures, one can see that the structures of the different types of wood were quite different. Pine wood had the largest pores, all ~15 µm. Birch wood exhibited a bimodal pore size distribution. A smaller peak was located at ~6 µm and the higher peak was at ~0.8 µm. Oak wood had three groups of pores, developed at 0.08, 0.22, and 6.9 µm. The structures of all pellets appeared to be similar, regardless of the origin of the initial material. The dominant pores in all pellets developed at ~7.9 µm and the second, smaller peak was located at ~1.2 µm. This uniformisation of the pellet structure may have been due to several factors: damage to the wood structure during sawing, the produced sawdust having a similar grain shape regardless of the initial material (as seen in the SEM images of the sawdust of different wood presented below), and the same conditions of the pelletisation process. This similarity of the pellet structure certified the application of the unified model for all pellets together in the further modelling stage.

Figure 4 presents SEM images of the surfaces of the cross-sections of wood blocks, sawdust particles, and broken pellets. Differences in the internal structures of the wood blocks, sawdust particles, and pellets can be easily observed. The surface structure of sawdust particles was more complex compared to that of wood blocks as a result of sawing. Pores between sawdust particles and the majority of their surface pores disappeared during the pelletisation process. On the surfaces of the broken pellets, some bonds between sawdust particles could be identified. Their sizes were markedly smaller than the mean size of the particles. As the size of the bond zone was evaluated to be ~40 μm, the bond radius of 20 μm was used as the baseline input data for calibrations of the DEM modelling parameters.

The compaction pressure–piston displacement relationships (*σ**_z_* versus Δ*h*) for pelletisation are presented in Figure 5a. All dependencies were similar. The noticeable increase in the compaction pressure started at a piston displacement of >10 mm (i.e., >50% of the total displacement). For a higher *MC*, the maximum compaction pressure was reached at a considerably lower piston displacement (except with oak) and the unloading process lasted longer (Figure 5a inset). This resulted in a lower bulk density of pellets produced from sawdust with a higher *MC*.

The stress–deformation dependencies for the studied pellets are depicted in Figure 5b. Δ*L* is the displacement of the loading piston and *D* is the pellet diameter measured directly before diametral compression. All dependencies of *σ*_1_ on Δ*L*/*D* were smooth and round without any sudden drops, which is typical for the ductile breakage mode.

The SD of the mean values of the tensile strength *σ**_f_* and the deformation at failure, Δ*L_f_*/*D*, indicate the degree of variability of the experimental relationships resulting from the natural differentiation of pellets (Table 3). Three groups of pellets can be distinguished with respect to the tensile strength: *σ**_f_* > 1 MPa (oak, *σ**_z_* = 120 MPa, and *MC* = 8%); *σ**_f_* ~ 0.5 MPa (oak, *σ**_z_* = 60 MPa, and *MC* = 8%; oak, *σ**_z_* = 120 MPa, and *MC* = 20%; pine, *σ**_z_* = 120 MPa, and *MC* = 8%); and *σ**_f_* < 0.3 MPa (all others).

An increase in *MC* and a decrease in *σ**_z_* resulted in a decrease in the tensile strength of the pellets. The highest tensile strength was observed for oak pellets (*MC* = 8% and *σ**_z_* = 120 MPa) and the lowest for birch pellets (*MC* = 20% and *σ**_z_* = 60 MPa). An increase in *MC* and decrease in the compaction pressure resulted in a fourfold decrease in the tensile strength of oak and birch pellets and an eightfold decrease for pine pellets. The range of the determined values of *σ**_f_* is consistent with the findings of other researchers for pellets produced from untreated wood materials [4,6,22]. The range of the deformation at failure, Δ*L_f_*/*D*, found in the present study (0.56–0.82) is very similar to that obtained by Gilvari et al. [40] in the diametral compression tests of biomass pellets (0.05–0.07).

## 3. Modelling

### 3.1. DEM Setup

#### 3.1.1. Contact Model

The linear hysteretic spring contact model introduced by Walton and Brown [45] was used for the simulations. The model was extended to account for linear adhesion in the form introduced by Luding [46] to keep particles glued and to minimise changes in the structure of the pellet during unloading and removal from the mould. The contact force–displacement scheme in the normal direction shown in Figure 6 accounts for the plastic contact deformation, elastic unloading and reloading, and attractive adhesion force:(5)$$f_{n}=\left\{\begin{array}{lll} k_{1}\delta_{n}, & & k_{2}(\delta_{n}-\delta_{n,0})\geq k_{1}\delta_{n}\;(\mathrm{loading}),\\ k_{2}(\delta_{n}-\delta_{n,0}), & & k_{1}\delta_{n}>k_{2}(\delta_{n}-\delta_{n,0})>-k_{c}\delta_{n}\;(\mathrm{loading}-\mathrm{unloading}),\\ -k_{c}\delta_{n}, & & -k_{c}\delta_{n}\geq k_{2}(\delta_{n}-\delta_{n,0})\;(\mathrm{unloading}), \end{array}\right.$$
where $f_{n}^{}$ is the contact normal force, *k*_1_ is the loading (plastic) stiffness, *k*_2_ is the unloading (elastic) stiffness, *k_c_* is the adhesive stiffness, *δ**_n_* is the overlap in the normal direction, and *δ**_n,_*_0_ is the residual overlap during unloading when the force $f_{n}^{}$ changes from repulsive to attractive interaction. Adhesive interactions keep the overlap *δ**_n_* among the contacts within the pellet removed from the mould at a level very close to the residual overlap *δ**_n,_*_0_.

The plastic stiffness *k*_1_ is related to the yield strength *p_y_* of a particle [47],
(6)$$p_{y}=\frac{2E^{}}{\pi }\sqrt{\frac{\delta_{n,y}}{r}},$$
by the following formula [48]:(7)$$k_{1}=5r^{*}\min(p_{y,i},p_{y,j}),$$
where *E* is the Young’s modulus, *r* is the radius of a particle, *δ**_n,y_* is the yielding overlap, $r^{*}=r_{i}r_{j}/(r_{i}+r_{j})$ is the equivalent radius of the contacting particles, and *p_y,i_* and *p_y,j_* are the yield strengths of particles *i* and *j*, respectively.

Energy dissipation in the normal direction results from the difference between the loading and unloading stiffnesses. The elastic stiffness *k*_2_ for unloading and reloading is related to *k*_1_ through the restitution coefficient *e* as follows [48]:(8)$$e=\sqrt{\frac{k_{1}}{k_{2}}}.$$

The particle–particle force in the tangential direction, $f_{t}^{c}$, was updated incrementally as follows:(9)$$f_{t}^{}=\left\{\begin{array}{ccc} \begin{array}{l} {\left(f_{t}^{}\right)}_{0}+k_{t}\Delta \delta_{t}+f_{t}^{d},\\ \frac{\Delta \delta_{t}}{\left|\Delta \delta_{t}\right|}\mu_{p-p}f_{n}^{}, \end{array} & \begin{array}{c} \mathrm{if}\\ \begin{array}{l} \mathrm{if} \end{array} \end{array} & \begin{array}{c} \left|f_{t}^{}\right|<\mu_{p-p}f_{n}^{},\\ \left|f_{t}^{}\right|\geq \mu_{p-p}f_{n}^{}, \end{array} \end{array}\right.$$
where ${\left(f_{t}^{}\right)}_{0}$ is the tangential force at the end of the previous timestep, *k_t_* and *δ**_t_* are the stiffness and overlap in the tangential direction, respectively, and *μ**_p-p_* is the particle–particle friction coefficient.

The stiffness in the tangential direction, *k_t_*, was assumed to be equal to the stiffness in the normal direction. The velocity-dependent dissipative component $f_{t}^{d}$ of the tangential force *f_t_* is defined as [48,49]:(10)$$f_{t}^{d}=-\sqrt{\frac{4m^{*}k_{t}}{1+{\left(\frac{\pi }{\ln e}\right)}^{2}}}v_{t},$$
where $m^{*}=m_{i}m_{j}/(m_{i}+m_{j})$ is the equivalent mass of the contacting particles, *v_t_* is the relative velocity in the tangential direction, and *e* is the same restitution coefficient used for the energy dissipation in the normal direction.

The dissipative torque *M**_i_* associated with rolling friction *m_r_* was introduced as
(11)$$M_{i}=-m_{r}f_{n}r_{i}\omega_{i},$$
where *ω_i_* is the unit angular velocity vector of particle *i* at the contact point.

The binding mechanism inside pellets is mainly based on the re-solidification of melted lignin during cooling [2]. Therefore, to bring the DEM modelling closer to the experimental conditions, the adhesive interactions in the contacts of the relaxed pellet were replaced by the parallel BPM, as proposed by Potyondy and Cundall [35]. In contrast to the lack of force interactions in the tangent direction assumed in the adhesion model, the BPM provides stiffness in the normal and tangential directions. Additionally, substituting adhesive interactions with the BPM protects the simulations against the appearance of secondary adhesive contacts during deformation, which might falsify the resultant breakage strength of the pellets. The BPM was initiated when the total kinetic energy of the assembly decreased below 10^−8^ J and the overlap was very close to the residual overlap, *δ*~*δ**_n_*_,0_ (Figure 7).

The forces and moments of the bond connections between the two particles were calculated incrementally [48] as follows:(12)$$\begin{array}{l} \Delta f_{n}^{b}=-v_{n}k_{n}^{b}A\Delta t,\\ \Delta f_{t}^{b}=-v_{t}k_{t}^{b}A\Delta t,\\ \Delta M_{n}^{b}=-\omega_{n}k_{t}^{b}J\Delta t,\\ \Delta M_{t}^{b}=-\omega_{t}k_{n}^{b}\frac{J}{2}\Delta t, \end{array}$$
where *v_n_* and *v_t_* are the relative velocities in the normal and tangential directions, respectively; $k_{n}^{b}$ and $k_{t}^{b}$ are the stiffnesses in the normal and tangential directions, respectively; $A=\pi r_{b}^{2}$ and $J=\frac{\pi r_{b}^{4}}{2}$ are the area and moment of inertia of the bond cross-section, respectively; *r_b_* is the radius of the bond; and Δ*t* is the time increment.

The Young’s modulus of the bond is:(13)$$E^{b}=k_{n}^{b}(r_{i}+r_{j}).$$

The bond is broken when the maximum normal stress $\sigma_{n,\max}^{b}$ exceeds the tension strength *σ**_c_* or the maximum tangent stress $\tau_{\max}^{b}$ exceeds the shear strength *τ**_c_*:(14)$$\begin{array}{l} \sigma_{n,\max}^{b}=\frac{-f_{n}^{b}}{A}+\frac{2M_{t}^{b}}{J}r_{b}>\sigma_{c},\\ \tau_{\max}^{b}=\frac{-f_{t}^{b}}{A}+\frac{M_{n}^{b}}{J}r_{b}>\tau_{c}. \end{array}$$

The critical bond force associated with the tension strength *σ**_c_* is
(15)$$f_{c}=A\sigma_{c},$$
as is depicted in Figure 7.

#### 3.1.2. DEM Modelling Parameters

Spherical particles simulating sawdust used in the experiments, with radii normally distributed in the range of 0.14–0.26 mm with the mean = 0.2 mm and SD = 0.02 mm, were generated randomly in a mould of the same size as the mould used in the experiments. The assembly consisted of 40,000 particles. The spherical shape of the particles was used to simplify and fasten the numerical calculations. The DEM simulation parameters are listed in Table 4.

The porosity of random bedding of spherical particles in the DEM simulations was ~37%. The high porosity of the bulk sawdust, ~80%, resulted from the superposition of two components: free spaces between particles and micro- and macro-pores on the surface of sawdust particles and (eventually) inside. Therefore, the porosity of the bedding of spherical particles could represent only the pores located between the sawdust particles. The internal pores inside the particles and pores resulting from the surface roughness of the sawdust particles were not covered by this approach. To model the compaction of deformable wood chips in a cycling loading test using the DEM, Xia et al. [38] set the bulk density of the intact wood as the solid density of particles and reproduced the compaction process well. Similarly, we wanted to set the bulk density of the intact wood as the solid density of the particles. However, we decided to set a particle solid density of 680 kg m^−3^ as the uniform value for the solid density of all types of sawdust particles. This value was slightly higher than the bulk density of intact oak wood and higher than those of pine and birch. We decided to set the same particle densities for each type of sawdust because we surmised that the structure of the sawdust of all types of wood would be uniform during sawing. Almost identical values of solid-phase density of each type of sawdust (see Table 1), along with almost identical pore size distributions of all pellets (see Figure 4), confirmed this assumption. The assumed high sawdust particle density was a direct consequence of the high solid-phase density. By setting the bulk density of the intact wood as the solid density of particles, in the first stage of compaction, the free spaces between particles were reduced to zero, and, with further compaction, an increase in the density of the pellet above the density of the intact wood reproduced the closing of internal micro- and macro-pores inside sawdust particles.

We set the coefficients of particle–particle friction, *μ_p-p_*, and particle–wall friction, *μ_p-w_*, to 0.5 and 0.15, respectively, in accordance with similar studies on biomass pellets [40] and pinewood chips [38]. The effect of the rolling friction coefficient *m_r_* on the behaviour of pellets was not considered because, in reality, the rolling of elongated dust particles seems to be absent. Therefore, a very low default value of *m_r_* = 0.01 was used in the simulations, as recommended by the EDEM software package [48]. A Poisson ratio of *ν* = 0.35 and a coefficient of restitution of *e* = 0.5 were adopted for the DEM simulations, as typical values for wood materials [7,30].

The modulus of elasticity of oak, *E* = 15.7 GPa [7], was set as the representative value of the modulus of elasticity of sawdust particles. The yield strength *p_y_* and the resulting plastic stiffness *k*_1_ were determined as the values that best fit the *σ**_z_*(Δ*h*/*h*_0_) relationship during the compaction process. The bond radius *r_b_* evaluated from the SEM images of the cross-sections of the pellets was used as baseline input data for the DEM simulations. The elastic properties of the bonds were treated as adjustable parameters for the calibration process. The bond elasticity modulus *E^b^*, the strength *σ**_c_*, and the final value of the bond radius *r_b_* were determined as the values providing the best fit of the *σ*_1_(Δ*L*/*D*) relationship during diametral compression.

To reduce the computational time, the particle density was increased by a factor of 10^4^. To keep the gravitational force unchanged, the gravitational acceleration was reduced by a factor of 10^4^. As also shown in [29], scaling the density by a factor of 10^6^ did not change the shape of the *σ*_1_(Δ*L/D*) characteristics and reduced the tensile strength *σ**_f_* by only 1.2% of the strength of the sample without density scaling. Therefore, scaling density by a factor of 10^4^ was considered to introduce an error of <1.2%. Time integration was performed with steps of 2 × 10^−6^ s, that is, 4% of the Rayleigh timestep [50].

#### 3.1.3. Stages of DEM Simulations

As illustrated in Figure 8, DEM simulations were performed in the following sequence: particle generation and filling of the mould (Figure 8a), compaction (Figure 8b), unloading and mould removal, relaxation (Figure 8c), BPM initiation, and then diametral compression (Figure 8d). Simulated sawdust particles were randomly generated in the space of the mould. After settlement, the particles were compacted with a virtual piston with an axial velocity of 3.3 × 10^−5^ m s^−1^ up to the assumed value of the compaction pressure *σ**_z_* of 60 or 120 MPa. The unloading process was performed in two stages: (1) unloading in the *z* direction by piston removal with the same velocity as during loading and (2) mould removal by increasing the mould radius. Next, after the assembly was relaxed, the BPM was initiated and the adhesive interactions were gradually reduced to zero. The last stage was the diametral compression between parallel plates with the same displacement rate as in the previous stages. The colours in Figure 8 represent the average compressive force exerted on the particles. The force scale ranges illustrate the compressive force differences in the simulated objects: the maximum compressive force in the relaxed pellet was five orders of magnitude lower than that in the compacted pellet (compare the scales “c” and “b” in Figure 8) and the maximum compressive force during diametral compression at *σ*_1_ = 0.7 MPa (~0.5*σ**_f_*) was one order of magnitude lower than that in the compacted pellet (scales “d” and “b” in Figure 8).

#### 3.1.4. Fit Quality

The quality of fit of the experimental stress–deformation relationship *σ*_1_(Δ*L*/*D*) obtained from the simulation was evaluated using the relative root mean-square error (*RRMSE*):(16)$$RRMSE=\sqrt{\frac{1}{n}\sum_{i=1}^{n}{\left(\frac{\sigma_{1,\exp}^{i}-\sigma_{1,DEM}^{i}}{\sigma_{1,\exp}^{i}}\right)}^{2}},$$
where $\sigma_{1,\exp}^{i}$ is the tension stress in the centre of a pellet (see Figure 1) obtained from the experiment, $\sigma_{1,DEM}^{i}$ is the tension stress obtained from the DEM simulation, and *n* is the number of samples in the *σ*_1_(Δ*L/D*) relationship.

#### 3.1.5. Calibration of Material Parameters for DEM Modelling

The impact of five material parameters of the DEM simulations was considered: the yield strength of particles *p_y_*, the coordination number *CN*, the bond radius *r_b_*, the bond elasticity modulus *E^b^*, and the bond tension and shear strength *σ**_c_* = *τ**_c_*. The values of *p_y_* and *CN* were calibrated against the experimental data of sawdust compaction and pellet relaxation. The bond radius was evaluated based on SEM images of broken pellets and a preliminary modelling of the diametral compression test fitting to the range of experimental values of tensile strength and deformation at failure. The elastic modulus and strength of the bonds were determined as a set of two independent parameters that provided the best fit to the experimental diametral compression stress–deformation relationship.

### 3.2. DEM Run

#### 3.2.1. Compaction of Sawdust: Plastic and Elastic Stiffness of Particles

The yield strength of sawdust particles *p_y_* was determined as the value providing the minimum *RRMSE* of the DEM approximation of the experimental stress–deformation relationship of compaction process. As presented in Figure 9, the experimental relationships between *σ**_z_* and Δ*h*/*h*_0_, where *h_0_* is the initial height of the sawdust in the mould, for compaction of sawdust were very similar for the three wood materials. Therefore, a common value of *p_y_* was applied for all three materials. The DEM simulation performed for *p_y_* = 100 MPa fitted all the experimental relationships very well for Δ*h*/*h*_0_ < 0.75 (*RRMSE* < 0.15). The ratio of the yielding overlap to the particle radius, *δ**_n,y_*/*r*, was 9.63 × 10^−3^. The plastic stiffness of particles, *k*_1_, corresponding to *p_y_* = 100 MPa, was 1 × 10^5^ N m^−1^, and the elastic stiffness, *k*_2_, was 4 × 10^5^ N m^−1^; consequently, the assumed value of the restitution coefficient was 0.5. The lower quality of approximation of the experimental relationship for the highest level of compaction (Δ*h*/*h*_0_ > 0.75) resulted from the difference in curvature between the experimental and simulated lines. It is possible that a nonlinear contact model could fit the experimental data better.

#### 3.2.2. Coordination Number

Figure 10 illustrates the DEM-simulated relationship of the coordination number *CN* versus the bulk density ratio of the pellet to the intact wood, *BD_p_*/*BD_w_*, during compaction. The coordination number *CN* increased nonlinearly with increasing *BD_p_*/*BD_w_*, being faster and nonlinear for *BD_p_*/*BD_w_* < 1 (as free spaces closed between particles) and slower and almost linear for *BD_p_*/*BD_w_* > 1 (as there remained no free spaces between particles, the surface pores on the sawdust particles and intraparticle pores being closed). In this figure, three values of *CN* of relaxed pellets (12.9, 12.2, and 10.5) were obtained in the process of unloading and relaxation of the bulk of particles compacted to *σ**_z_* = 120, 60, and 48 MPa, respectively. The values of the *BD_p_*/*BD_w_* ratio, corresponding to the indicated values of *CN*, respectively, covered the entire range of variability of the experimental values of the *BD_p_*/*BD_w_* ratio of 1.62 ± 0.19, 1.32 ± 0.14, and 1.13 ± 0.08, determined as the averaged values for the three wood materials (Table 2) and for three cases: (1) *MC* = 8%, *σ**_z_* = 120 MPa; (2) *MC* = 8%, *σ**_z_* = 60 MPa and *MC* = 20%, *σ**_z_* = 120 MPa; and (3) *MC* = 20%, *σ**_z_* =120 MPa.

The diametral compression tests were simulated for the three selected values of the coordination number of the pellets (Figure 11a). The shapes of the stress–deformation relationships changed slightly with the change in *CN*. All shapes were typical for ductile behaviour with progressive loading and gradual collapse. The tensile strength *σ**_f_* increased slightly faster than linearly as the coordination number increased (Figure 11b). However, Gilvari et al. [40] found that the tensile strength of pellets increased linearly with the coordination number.

#### 3.2.3. Bond Radius Estimation

As stated previously, the mean size of bonds of 20 μm was the baseline input data for searching for the bond radius in the DEM simulations. Examples of simulations of the diametral compression performed for different values of the bond radius and constant values of the bond elasticity modulus and bond strength (*E^b^* = 120 MPa and *σ**_c_* = 36 MPa) illustrate the strong dependence of the stress–deformation relationships on the bond radius (Figure 12a). However, simulations performed for the bond parameters fulfilling the conditions $E^{b}/r_{b}^{2}=\mathrm{constant}$ and $E^{b}/\sigma_{c}=\mathrm{constant}$ had almost identical stress–deformation relationships (Figure 12b). This similarity means that the model can be easily recalibrated to different values of micro-parameters to simulate other stages of compaction (coordination number), other bond radii, or other materials (elastic parameters). This indicates an important need for precise experimental verification of the micro-parameters of bonds to obtain reliable modelling results.

Simulations performed with constant values of *E^b^* and *σ**_c_* and variable bond radii indicated that the breakage strength *σ**_f_* (Figure 13a) and the deformation at failure, Δ*L_f_*/*D* (Figure 13b), increased almost linearly with the bond cross-sectional area *A*. Experimental values of *σ**_f_* and Δ*L_f_*/*D* for oak pellets (*MC* = 8% and 20% and *σ**_z_* = 60, 120 MPa) presented in this figure served as reference data to find the best-fitting values of *E^b^* and *σ**_c_* for the selected value of *r_b_* of 20 μm and the particular values of *MC* and *σ**_z_*. By taking into account the fact that the value of the bond radius approximated from SEM images provides a similarity between the simulated and experimental stress–deformation relationships and the breakage profiles, a bond radius of 20 μm was set as the fixed material parameter for all further DEM simulations.

#### 3.2.4. Bond Elasticity and Strength

The bond elasticity modulus *E^b^* and strength *σ**_c_* were determined as values providing the minimum *RRMSE* of the DEM approximation of the experimental stress–deformation relationship. Details of the procedure for searching for the best-fitting parameters are exemplarily illustrated for oak pellets with *MC* = 8% compacted to *σ**_z_* = 120 MPa (Figure 14). Generally, 8–10 simulations, divided into two or three groups with sequentially changed *E^b^* (Figure 14a) and *σ**_c_* (Figure 14b) as the constant and variable parameters, were sufficient to find the set of *E^b^* and *σ**_c_*, bringing the *RRMSE* relatively close to its global minimum. This procedure was applied separately to each variant of the experimental relationship. Eighty simulations were performed to find the best fit for all experimental cases.

## 4. Simulation Results

Figure 15 presents examples of experimental relationships between the tension stress and the diametral deformation *σ*_1_(Δ*L*/*D*) fitted by the DEM approximations. The DEM simulations approximated very well the individual experimental stress–deformation relationships that were the closest to the mean tensile strength (*σ**_f_*, Δ*L_f_*/*D*).

The best fit with the experimental relationships was obtained for *CN* corresponding to the degree of compaction of pellets, as given in Table 5.

The best-fitting *CN* increased with increasing *BD_p_*/*BD_w_* ratio. This can be clearly seen for the oak pellets. The bond elasticity modulus *E^b^* and its strength *σ**_c_* decreased as *MC* increased and *σ**_z_* decreased. The range of variability of the bond elasticity modulus applied in the DEM simulations (Table 5) from 9.2 MPa (birch, *MC* = 20%, and *σ**_z_* = 60 MPa) to 120 MPa (oak, *MC* = 8%, and *σ**_z_* = 120 MPa) was consistent with a range of values (3–146 MPa) determined experimentally for wood pellets in the puncture test [51,52,53]. The approximately twofold decrease in the bond elasticity modulus with an increase in *MC* from 8% to 20% applied in our simulations fitted very well to the rate of decrease in pellet elasticity determined experimentally by Gallego et al. [51]. The order of magnitude of the bond elasticity modulus (10^7^–10^8^ Pa) and the corresponding range of the normal stiffness coefficient of the bond (10^10^–10^11^ N m^−3^) applied in our study were similar to the values of the bond elasticity and stiffness applied in the DEM modelling of loading of pinewood chip briquettes by Xia et al. [38], the durability of wood pellets found by Mahajan et al. [39], and the breakage of biomass pellets studied by Gilvari et al. [40]. The maximum bond tensile strength *σ**_c_* of 36 MPa applied in our study for oak (*MC* = 8% and *σ**_z_* = 120 MPa) corresponded to that applied by Gilvari et al. [40] (*σ**_c_* = 35 MPa) and Mahajan et al. [39] (*σ**_c_* = 40 MPa) to model pellet behaviour. The ~30% decrease in the bond strength with an increase in *MC* from 8% to 20% was consistent with the results of Whittaker and Shield [2] and Li and Liu [9], who indicated that an excessively high *MC* reduces the binding forces between particles.

The high value of the coordination number applied in simulations (10.5–12.9) considerably increased the strength of pellets generated by elastic bonds and friction in particle contacts and promoted the rounded shape of the *σ*_1_(Δ*L*/*D*) relationship typical for the ductile breakage mode. As shown by Horabik et al. [54], *σ**_c_*/*E^b^* was <0.1 for the semi-brittle breakage mode and >0.15 for ductile breakage. The *σ**_c_*/*E^b^* values found in this study ranged from 0.25 to 0.5, which is typical for the ductile breakage mode.

Figure 16 presents a comparison of experimental and simulated patterns of cracks of oak pellets.

Cracks were initiated on the flat surfaces of pellets, and progressive deformation developed along the direction of loading toward the interior of the pellet. Two types of breakage patterns were observed in the experiments: a single crack slightly inclined against the direction of loading and diffuse small cracks connected into branches. The first type occurred for pellets of the highest and the lowest strength and the second for the intermediate strengths. The DEM simulations indicated that the breakage profile was strongly influenced by the coordination number and was very slightly influenced by the elastic parameters of the bonds. For *CN* = 12.9, the DEM simulations reproduced a single-crack breakage well. Simulations performed for *CN* = 12.2 reproduced the development of diffuse cracks. In general, cracks grew in the central part of the pellet and did not reach the outer surface of the pellet. The earliest appearing cracks were observed in simulations performed for *CN* = 10.5. In the first stage, breakage occurred as a set of small, separated cracks. As the deformation progressed, these cracks gradually merged into a single zone, which finally separated the pellet into two parts.

## 5. General Remarks

Bonding mechanisms that join sawdust particles are crucial for the mechanical strength of pellets. The diametral compression test provides a convenient method for determining the tensile strength of pellets. The DEM is a powerful tool for investigating the influence of particle-scale properties on bulk material behaviour. Bonded spheres applied in the DEM modelling proved to be useful for reproducing the breakage behaviour of wood pellets. The main novelty of this research was the good quality of the DEM reproduction of particular phases of pellet formation and their breakage behaviour during diametral compression. Pellet formation was reproduced with acceptable quality, while the response of pellets to diametral compression was reproduced with high accuracy (*RRMSE* < 0.12). The DEM simulations reproduced the shape of the stress–deformation relationships, tensile strength, and breakage profiles of pellets produced from all three wood materials at two levels of *MC* and at two levels of compaction pressure. The order of magnitude of the elastic modulus and the strength of bonds were consistent with previously published research results on the application of the BPM for testing the breakage strength of wood pellets [40].

The range of deformation of particles, which was necessary to model the behaviour of sawdust particles during the pelletisation process, was far beyond the typical range of deformation considered in the DEM simulations. To overcome this limitation, a uniform and high density similar to the bulk density of the intact oak wood was set as the bulk density of all sawdust particles (regardless of the type of wood). Similarly, Xia et al. [38] used wood bulk density as the particle density in their DEM simulations. By this means, the large bulk deformation of sawdust particles during pelletisation was considered to result from the reduction of free spaces between particles and compaction of internal pores inside sawdust particles reproduced by a very high level of overlap (*δ**_n_*/*r*~0.33). The coordination number of relaxed pellets in the range of 10.5–12.9 was consistent with the values obtained by Nordström et al. [27] in their DEM modelling of the formation of microcrystalline cellulose tablets under pressure in the range of 100–300 MPa.

To model the stress–deformation relationship during diametral compression of pellets with a bond radius equal to the radius of contact between particles, the stiffness and strength of the bond should be unrealistically low compared to the experimental findings [50]. It is reasonable to expect that bonding develops only on a fraction of the contact area of the largely deformed sawdust particles. Therefore, supported by the results of the SEM images of the pellets and the results of the calibration procedure, the bond radius applied to the DEM simulations was assumed to be a small fraction (~0.16) of the contact radius. The assumed value of the initial particle density, along with placing the bond radius as a small fraction of the contact radius, allowed the proper modelling of both pelletisation and diametral compression. The simulations yielded good qualitative agreement of the crack profile with the experimental data, reproduction of the stress–deformation relationship during diametral compression, and agreement of the values of the simulated bond elasticity with the experimental data.

Some problems encountered during the DEM modelling of sawdust and their products may have resulted from very limited access to verified experimental data for the micro-properties of sawdust particles [51]. Therefore, more experimental data for micro-properties are necessary to facilitate further development of DEM modelling of these materials.

## 6. Conclusions

An experimental study and numerical simulations were performed for pressure compaction of sawdust and diametral compression tests of sawdust pellets. The simulation results reproduced the results of laboratory testing well and provided deeper insight into particle–particle bonding mechanisms. The main novelty of the present study was the successful simulation of the diametral compression test of wood pellets and reproduction of the profiles of ductile crack formation using the DEM with the implemented BPM. The following can be concluded from this study:The stress–deformation relationship during pelletisation of wood sawdust comprising a large range of particle deformations was successfully modelled using a DEM equipped with a linear hysteretic contact model. Good agreement between the simulated range of change in the bulk density during pelletisation with experimental data was obtained because of the application density of the intact wood as the input density of particles in the DEM simulations;The highest tensile strength was obtained for oak and the lowest for birch pellets. For all materials, the tensile strength was the highest for *MC* = 8% of sawdust compacted under a pressure of 120 MPa and the lowest for *MC* = 20% of sawdust compacted under a pressure of 60 MPa;The breakage processes of pellets of all tested materials were successfully simulated using the DEM with the BPM;All pellets exhibited a ductile breakage mode characterised by a smooth and round stress–deformation relationship without any sudden drops. Cracks were initiated in locations close to the centre of the pellet, and progressive deformation developed in the direction of loading and toward the interior of the pellet;Applying values of the bond elasticity modulus *E^b^* and the tensile strength *σ*_c_ fulfilling the condition *σ*_c_/*E^b^* > 0.25 in the DEM simulations allowed the stress–deformation relationship and crack formation to be reproduced well for all studied pellets.

## Figures and Tables

**Figure 1 materials-14-03273-f001:**
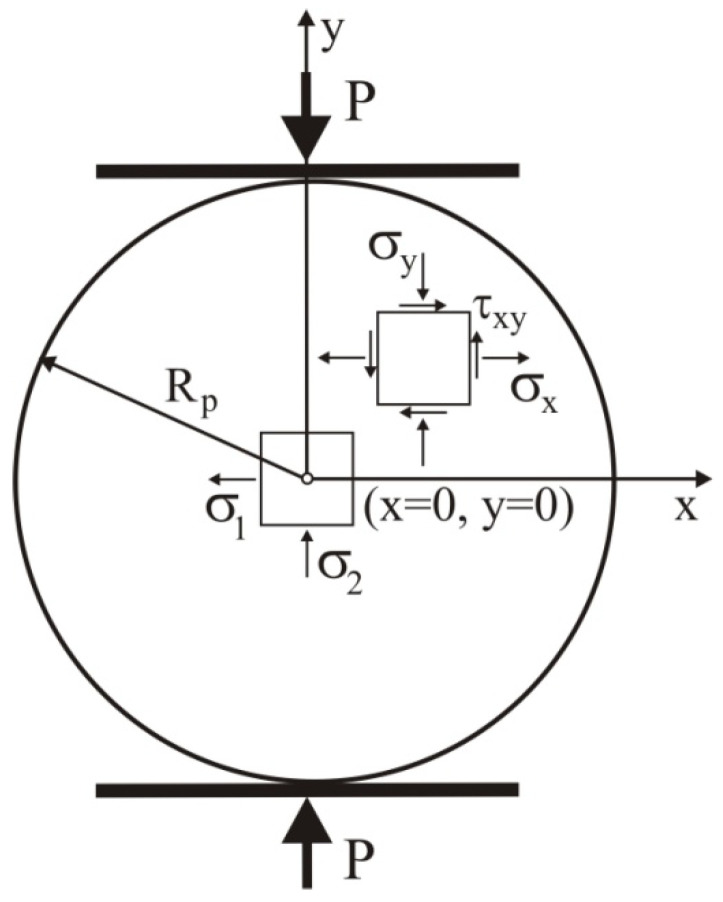
Stress components in the pellet during the diametral compression test.

**Figure 2 materials-14-03273-f002:**
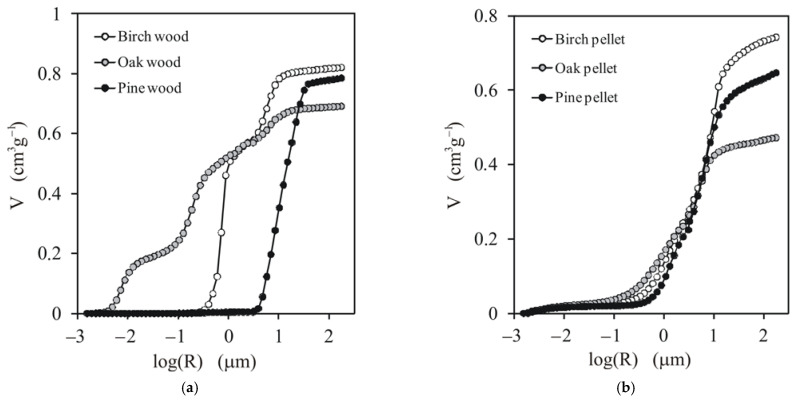
Dependencies of the pore volume on the pore radius for (**a**) the studied wood and (**b**) pellets. Average curves from two replicates are presented.

**Figure 3 materials-14-03273-f003:**
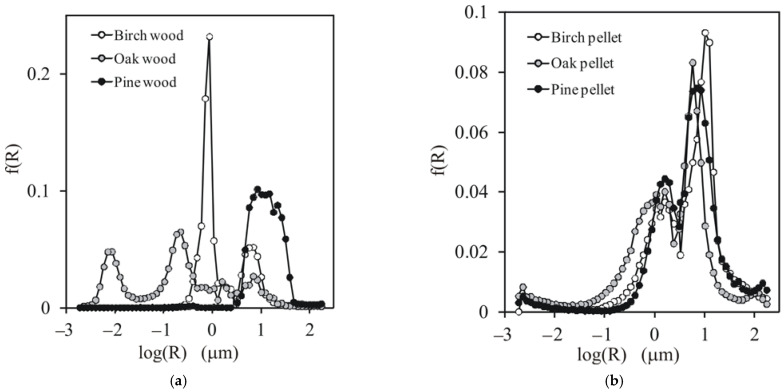
Pore size distribution functions for (**a**) the studied wood and (**b**) pellets. Average curves from two replicates are presented.

**Figure 4 materials-14-03273-f004:**
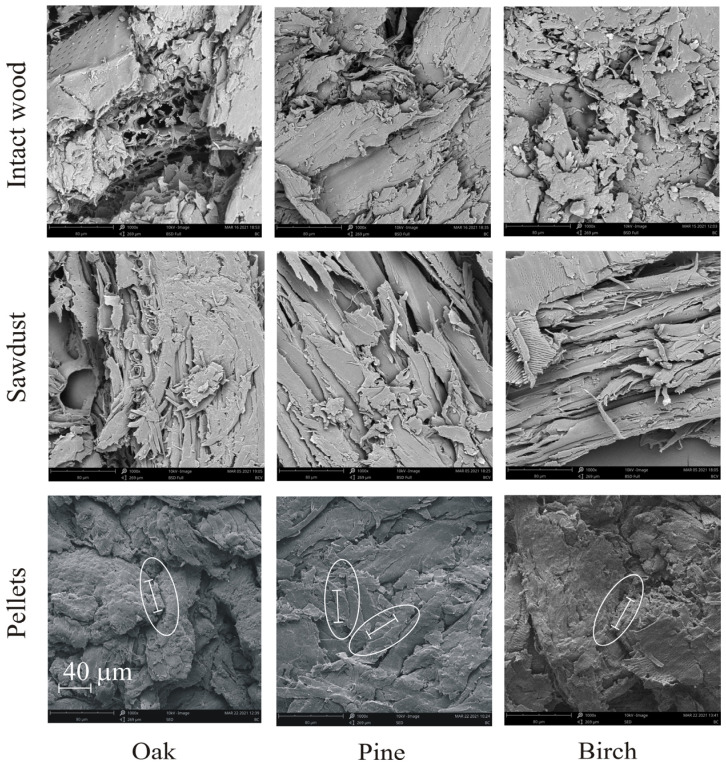
Representative SEM images (at 1000× magnification) of the surfaces of cross-sections of the wood blocks, sawdust particles, and pellets. White ellipses surround examples of potential particle–particle bonds. The 40 µm scale bar shown in the oak pellet image is valid for all other materials.

**Figure 5 materials-14-03273-f005:**
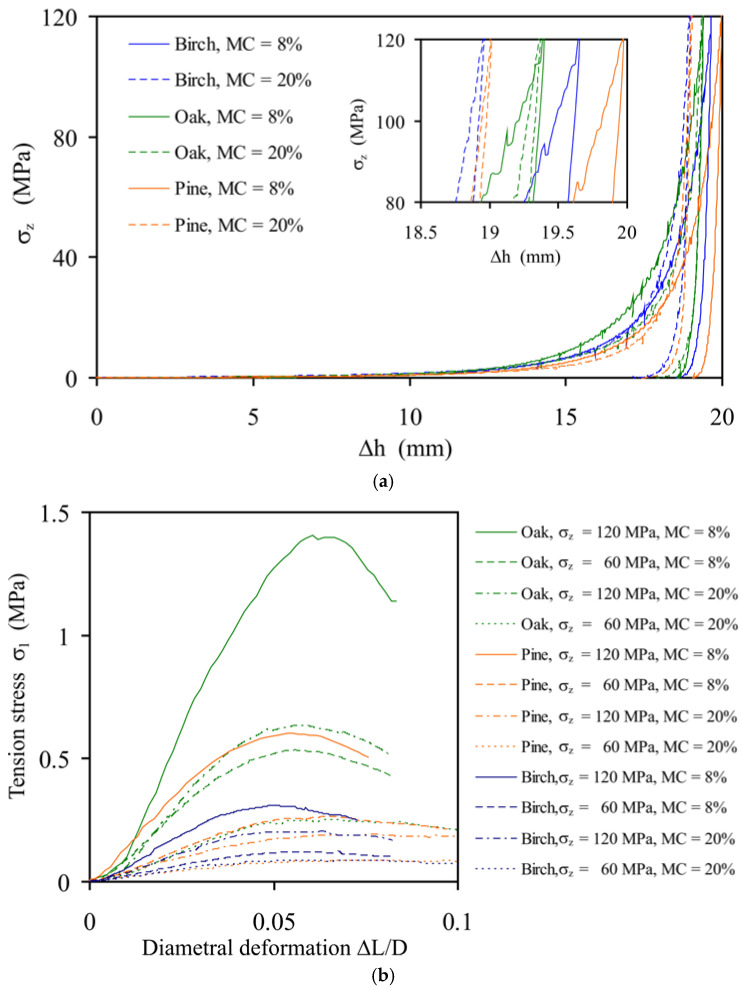
Stress–deformation dependencies during (**a**) the compaction–unloading cycle of sawdust and (**b**) the diametral compression of pellets.

**Figure 6 materials-14-03273-f006:**
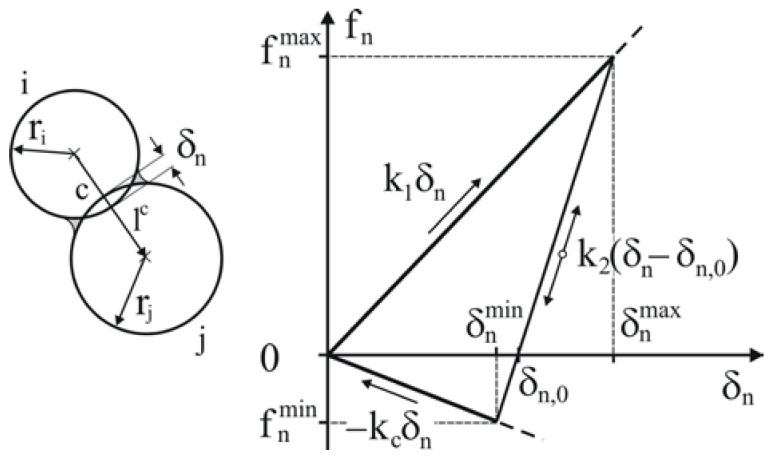
Linear hysteretic contact model with adhesion.

**Figure 7 materials-14-03273-f007:**
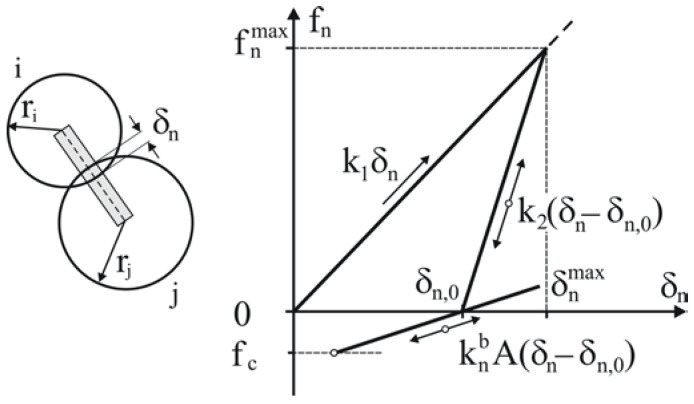
Linear hysteretic contact model with the BPM.

**Figure 8 materials-14-03273-f008:**
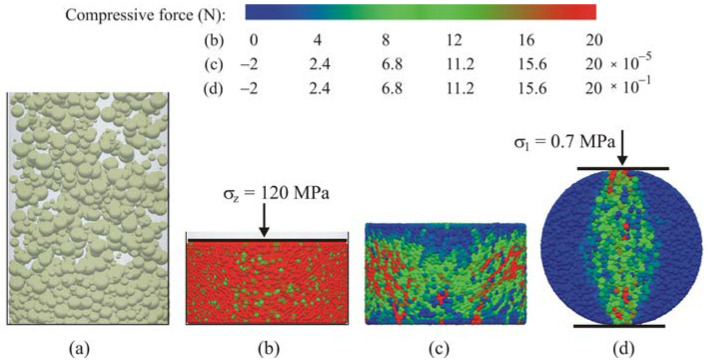
Stages of the DEM simulations: (**a**) filling, (**b**) compaction, (**c**) relaxation, and (**d**) diametral compression.

**Figure 9 materials-14-03273-f009:**
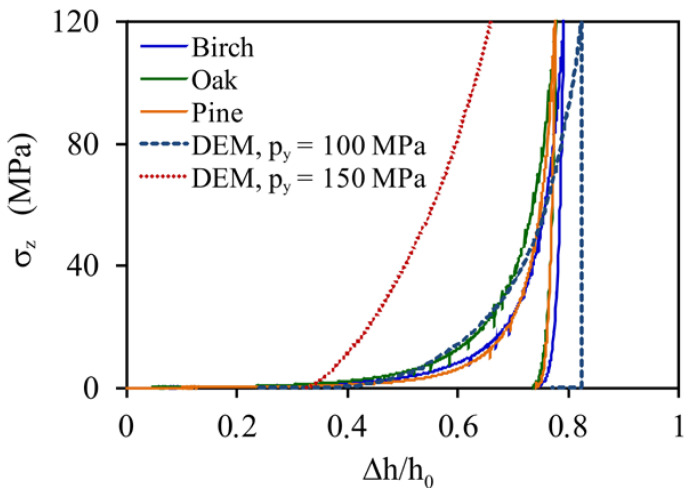
Fitting the compaction and unloading cycle *σ**_z_*(Δ*h*/*h*_0_) of sawdust.

**Figure 10 materials-14-03273-f010:**
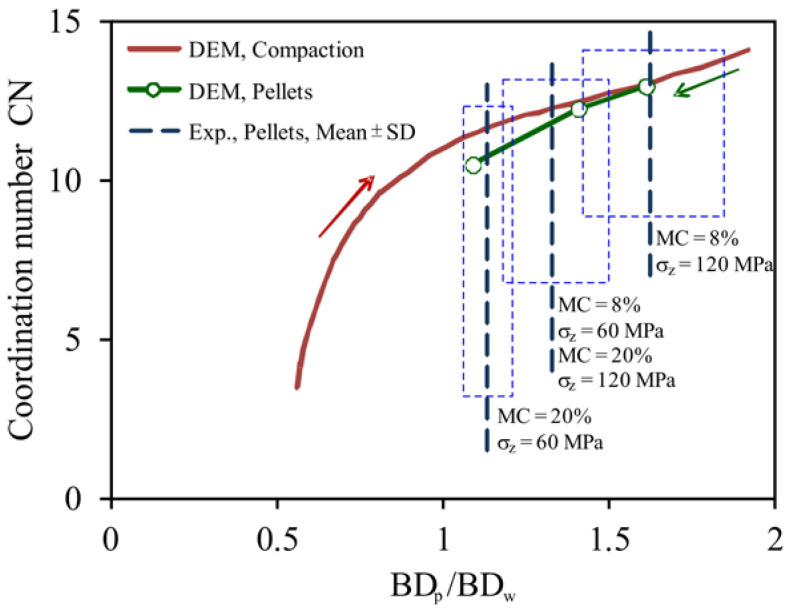
Comparison of the coordination number CN versus the bulk density ratio *BD_p_*/*BD_w_* of sawdust during compaction and for relaxed pellets.

**Figure 11 materials-14-03273-f011:**
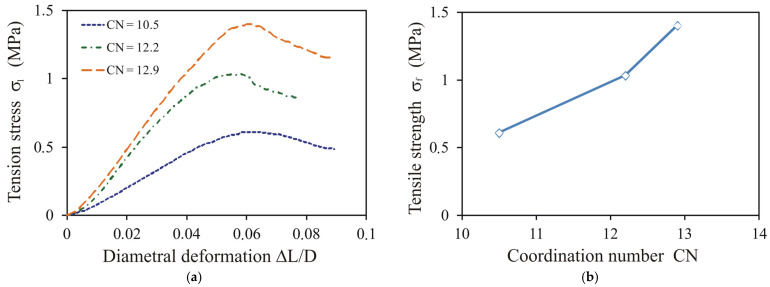
Impact of the coordination number *CN* on (**a**) the shape of the stress–deformation relationship during diametral compression and (**b**) the tensile strength for *r_b_* = 20 μm, *E^b^* = 120 MPa, and *σ*_c_ = 36 MPa.

**Figure 12 materials-14-03273-f012:**
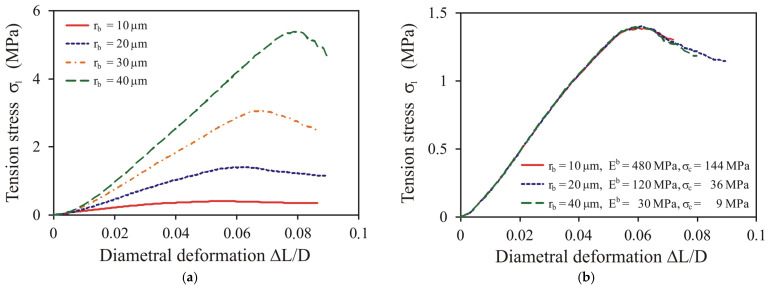
Stress–deformation relationship during diametral compression: (**a**) impact of the bond radius for *E^b^* = 120 MPa, *σ**_c_* = 36 MPa, and *CN* = 12.9; (**b**) impact of the bond radius compensated by the constant value of the following proportions: *E^b^*⋅*r_b_*^2^ = constant and *E^b^*/*σ**_c_* = constant.

**Figure 13 materials-14-03273-f013:**
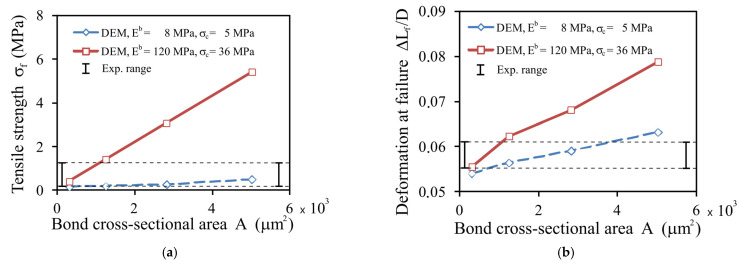
DEM simulations fitting the range of experimental values for oak as influenced by the bond cross-sectional area *A* for *CN* = 12.9: (**a**) the tensile strength, *σ**_f_*; (**b**) deformation at failure, Δ*L_f_/D*.

**Figure 14 materials-14-03273-f014:**
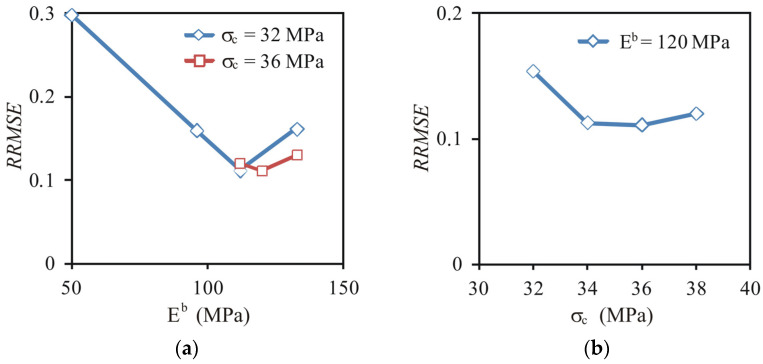
*RRMSE* of fitting the experimental stress–deformation relationship during diametral compression of oak pellets (*MC* = 8% and *σ*_z_ = 120 MPa) performed for *CN* = 12.9 and *r_b_* = 20 μm as influenced by (**a**) the bond elasticity modulus *E^b^* and (**b**) the bond strength *σ*_c_.

**Figure 15 materials-14-03273-f015:**
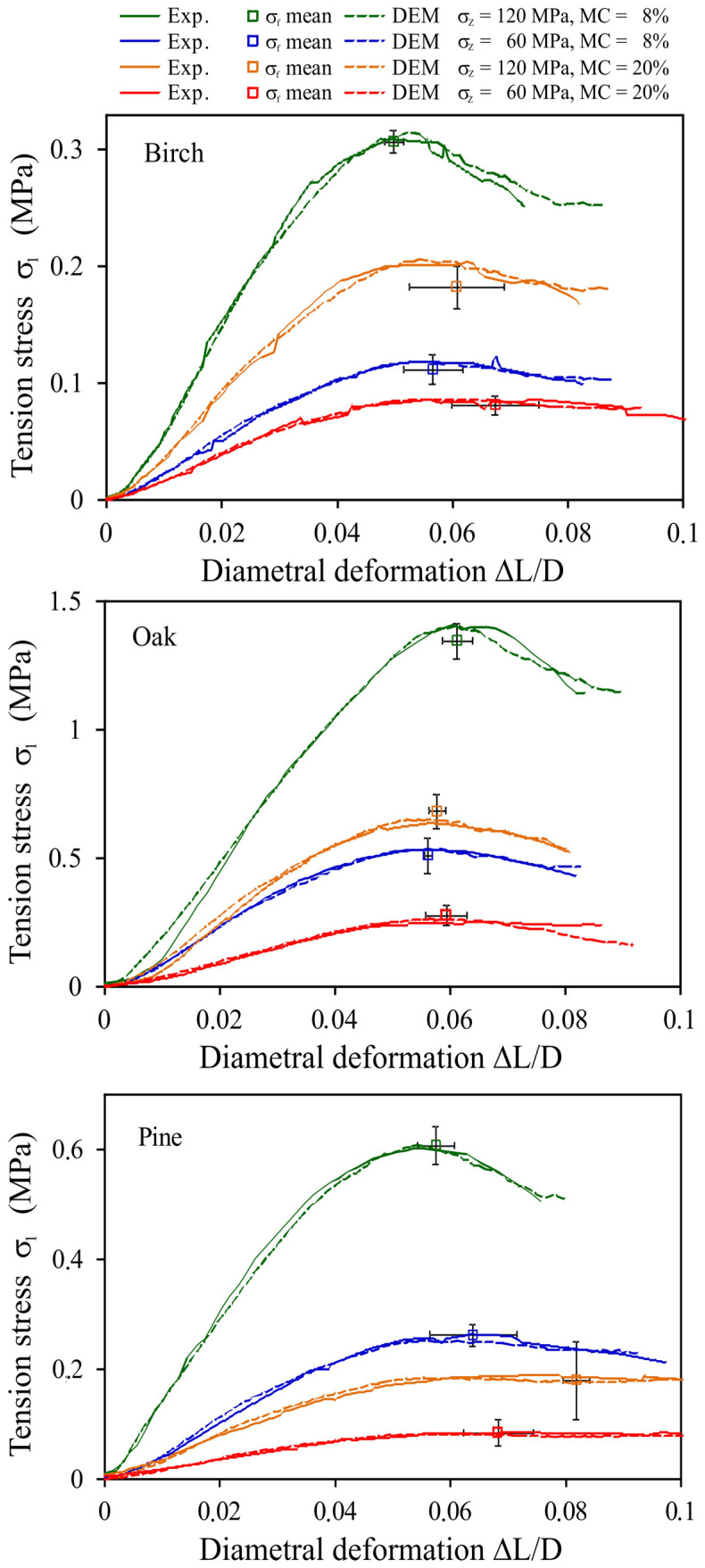
Fitting the *σ*_1_*(*Δ*L/D)* relationships of the diametral compression for birch, oak, and pine. The bars indicate the SD of the mean value.

**Figure 16 materials-14-03273-f016:**
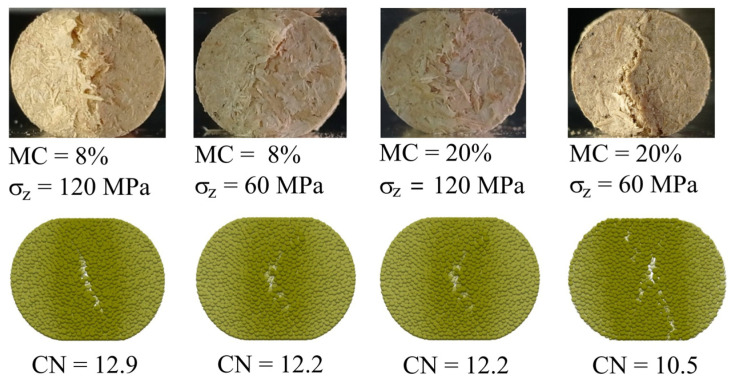
Experimental and simulated breakage profiles of oak pellets for Δ*L/D* = 0.1.

**Table 1 materials-14-03273-t001:** Density of the solid phase of the intact wood and of the sawdust measured by helium pycnometry.

Material	Wood Solid Phase (Including Closed Pores)	Sawdust
	Density *ρ**_w_* (kg m^−3^)	Density *ρ**_s_* (kg m^−3^)
Birch	1369.8 ± 0.3 ^a^	1465.3 ± 0.8 ^a^
Oak	1068.4 ± 1.6 ^b^	1459.9 ± 1.1 ^b^
Pine	1426.6 ± 0.3 ^c^	1468.7 ± 0.6 ^c^

Mean values in a column followed by the same letter are not significantly different at the 5% level.

**Table 2 materials-14-03273-t002:** Bulk density and porosity of the studied intact wood, sawdust, and pellets at various values of *MC*.

	*MC* (%)	Intact Wood		Sawdust (Initial)		Pellets			
		*BD_w_* (kg m^−3^)	*p_w_* (%)	*BD_s_* (kg m^−3^)	*p_s_* (%)	Compacted to 60 MPa		Compacted to 120 MPa	
						*BD_p_* (kg m^−3^)	*p_p_* (%)	*BD_p_* (kg m^−3^)	*p_p_* (%)
Birch	8	550 ± 2 ^a^	62.5 ± 0.2 ^a^	301.7 ± 6.4 ^a^	79.4 ± 1.7 ^a^	701.2 ± 9.9 ^a^	52.1 ± 0.7 ^a^	843.9 ± 4.8 ^a^	42.4 ± 0.2 ^a^
Birch	20			272.1 ± 9.7 ^b^	81.4 ± 2.9 ^b^	634.6 ± 6.7 ^b^	56.7 ± 0.6 ^b^	781.9 ± 22 ^b^	46.6 ± 1.3 ^b^
Oak	8	675 ± 3 ^b^	53.8 ± 0.2 ^b^	312.6 ± 5.4 ^a^	78.6 ± 1.4 ^a^	824.5 ± 10.4 ^c^	43.5 ± 0.5 ^c^	1005.8 ± 11.8 ^c^	31.1 ± 0.4 ^c^
Oak	20			267.8 ± 6.3 ^b^	81.6 ± 1.9 ^b^	700.3 ± 25.5 ^a^	52.0 ± 1.9 ^a^	826.4 ± 35.5 ^a^	43.3 ± 1.9 ^a^
Pine	8	510 ± 2 ^c^	65.2 ± 0.2 ^c^	267.9 ± 6.7 ^b^	81.8 ± 2.0 ^b^	779.6 ± 12.9 ^d^	46.9 ± 0.8 ^d^	937.8 ± 17.8 ^d^	36.2 ± 0.7 ^d^
Pine	20			250.9 ± 3.4 ^c^	82.9 ± 1.1 ^c^	610.9 ± 44.2 ^ab^	58.4 ± 4.2 ^ab^	673.1 ± 33.3 ^b^	54.2 ± 2.7 ^e^

Mean values in a column followed by the same letter are not significantly different at the 5% level.

**Table 3 materials-14-03273-t003:** Tensile strength of pellets.

Material	*σ**_z_* (MPa)	*MC* (%)	Δ*L_f_*/*D*	*σ**_f_* (MPa)
Birch	60	8	0.056 ± 0.005 ^a^	0.112 ± 0.012 ^a^
Birch	120	8	0.056 ± 0.003 ^a^	0.307 ± 0.009 ^b^
Birch	60	20	0.057 ± 0.005 ^a^	0.081 ± 0.008 ^a^
Birch	120	20	0.061 ± 0.008 ^b^	0.183 ± 0.018 ^c^
Oak	60	8	0.056 ± 0.001 ^a^	0.511 ± 0.069 ^d^
Oak	120	8	0.061 ± 0.003 ^b^	1.346 ± 0.068 ^e^
Oak	60	20	0.059 ± 0.004 ^c^	0.277 ± 0.039 ^b^
Oak	120	20	0.058 ± 0.002 ^c^	0.683 ± 0.067 ^d^
Pine	60	8	0.064 ± 0.008 ^b^	0.262 ± 0.019 ^b^
Pine	120	8	0.057 ± 0.003 ^a^	0.608 ± 0.034 ^d^
Pine	60	20	0.068 ± 0.006 ^d^	0.085 ± 0.025 ^a^
Pine	120	20	0.082 ± 0.002 ^e^	0.182 ± 0.071 ^c^

Mean values in a column followed by the same letter are not significantly different at the 5% level.

**Table 4 materials-14-03273-t004:** DEM simulation parameters.

Parameter	Symbol	Value
Container:		
Diameter (mm)	*D_c_*	10
Height (mm)	*H_c_*	25
Solid density (kg m^–3^)	*ρ*	7800
Young’s modulus (MPa)	*E*	1.561 × 10^6^
Poisson’s ratio	ν	0.3
Particles:		
Number		40,000
Mean particle radius (mm)	*r*	0.2
SD of particle radius (mm)	*r_sd_*	0.02
Particle radius range (mm)		0.14–0.26
Particle solid density (kg m^–3^)	*ρ*	680
Young’s modulus (MPa)	*E*	1.57 × 10^4^
Poisson’s ratio	ν	0.35
Yield strength (MPa)	*p_y_*	100; 150
Mean loading (plastic) stiffness (N m^–1^)	*k* _1_	1 × 10^5^
Mean unloading (elastic) stiffness (N m^–1^)	*k* _2_	4 × 10^5^
Mean adhesion stiffness (N m^–1^)	*k_c_*	0; 600
Restitution coefficient	*e*	0.5
Particle–particle friction coefficient	*μ* *_p-p_*	0.5
Particle–wall friction coefficient	*μ* *_p-w_*	0.15
Rolling friction coefficient	*m_r_*	0.01
Bond radius (μm)	*r_b_*	10; 20; 30; 40
Bond tension strength (MPa)	*σ* *_c_*	3–40
Bond shear strength (MPa)	*τ* *_c_*	3–40
Bond Young’s modulus (MPa)	*E^b^*	5–120

**Table 5 materials-14-03273-t005:** Best-fitting parameters of the DEM simulations.

Material	*σ**_z_* (MPa)	*MC* (%)	*CN*	*E^b^* (MPa)	*σ*_c_ (MPa)	*RRMSE*
Birch	60	8	12.2	12.8	4.6	0.027
Birch	120	8	12.2	36.6	10.1	0.039
Birch	60	20	12.2	9.2	4	0.037
Birch	120	20	12.2	22.4	8.05	0.039
Oak	60	8	12.2	60	20	0.027
Oak	120	8	12.9	120	36	0.111
Oak	60	20	10.5	52.8	13	0.089
Oak	120	20	12.2	73.2	24	0.116
Pine	60	8	12.2	27.2	13.5	0.052
Pine	120	8	12.9	50.4	13.5	0.028
Pine	60	20	12.2	8.6	4.3	0.053
Pine	120	20	12.2	19.7	9.9	0.049

## Data Availability

The data are available from the corresponding author for reasonable requests.

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
