# Peer review of "Breakage Strength of Wood Sawdust Pellets: Measurements and Modelling"

_materials, 2021, doi:10.3390/ma14123273_

Round 1

Reviewer 1 Report

The article is interesting and nicely written describing the theoretical verification of the mechanical properties of pellets. The article is written in great detail, which makes it quite long and difficult to follow. For example, certain parts could be omitted, such as the theoretical part of DEM (chapter 3.1.1. Contact model), since this part is taken from the literature, as it appears from the text.

In addition, the article contains some data for which no sources are given, such as saccharose density (L162).

It is also not entirely clear what the sentence on L174-175 is supposed to mean. "Means iMean values in a column followed by the same letter are not significantly different at the 5% level."

Also in Table 2, the values are labeled with letters where the means of each group are supposed to be significantly different from each other - the current labeling is not clear because it is not apparent which means are significantly different from each other.

Author Response

Dear Reviewer,

Thank you very much for reviewing our manuscript and for your valuable comments, which helped us to improve the manuscript. All comments were considered. Introduced changes were highlighted in the manuscript.

The article is interesting and nicely written describing the theoretical verification of the mechanical properties of pellets. The article is written in great detail, which makes it quite long and difficult to follow. For example, certain parts could be omitted, such as the theoretical part of DEM (chapter 3.1.1. Contact model), since this part is taken from the literature, as it appears from the text.

We agree with the reviewer opinion that some parts of the manuscript are too detailed. However, to help the potential readers in quick access to differences between two contact models applied for the study (adhesive for compaction and unloading of sawdust and BPM for pellets) we decided to leave the description of both contact models with all details.

In addition, the article contains some data for which no sources are given, such as saccharose density (L162).

The following reference was added:

“PubChem, National Center for Biotechnology Information, U.S. National Library of Medicine. https://pubchem.ncbi.nlm.nih.gov/compound/5988#section=Density (Last visited 10.06.2021)”

It is also not entirely clear what the sentence on L174-175 is supposed to mean. "Means iMean values in a column followed by the same letter are not significantly different at the 5% level."

The footer of Table 1 was corrected as follows: “Means values in a column followed by the same letter are not significantly different at the 5% level.”

Also in Table 2, the values are labeled with letters where the means of each group are supposed to be significantly different from each other - the current labeling is not clear because it is not apparent which means are significantly different from each other.

Labeling concerns each column separately and starts with the first raw. Standard deviation was placed to provide additional information on scatter of data.

Reviewer 2 Report

The manuscript deals with measuring and modelling of the breakage of wood sawdust pellets. It is divided in six sections: Introduction, Experiments, Modelling, Simulation results, General remarks and Conclusions. The experimental part contains diametral (Brazilian disk type) compression tests of the pellets at different levels of moisture content and compaction pressures. Three different wood species: Birch, Oak and Pine are used. Pore size distribution functions are measured. Modelling has been performed with the Discrete Element Method (DEM) with linear hysteretic spring contact model including adhesion. Parametric studies have been performed, e.g. the effect of coordination number of packaging, bond radius, bond cross-sectional area etc. Mass scaling was used to speed up the computations.

The article is well written and structured and deserves publication. The authors may consider to describe the parameter estimation procedure in more detail. It is also recommended to revise the last paragraph of the section General remarks starting “Some problems….” Such vague expression do not belong in a scientific article.

Author Response

Dear Reviewer,

Thank you very much for reviewing our manuscript and for your valuable comments, which helped us to improve the manuscript. All comments were considered. Introduced changes were highlighted in the manuscript.

The manuscript deals with measuring and modelling of the breakage of wood sawdust pellets. It is divided in six sections: Introduction, Experiments, Modelling, Simulation results, General remarks and Conclusions. The experimental part contains diametral (Brazilian disk type) compression tests of the pellets at different levels of moisture content and compaction pressures. Three different wood species: Birch, Oak and Pine are used. Pore size distribution functions are measured. Modelling has been performed with the Discrete Element Method (DEM) with linear hysteretic spring contact model including adhesion. Parametric studies have been performed, e.g. the effect of coordination number of packaging, bond radius, bond cross-sectional area etc. Mass scaling was used to speed up the computations.

The article is well written and structured and deserves publication. The authors may consider to describe the parameter estimation procedure in more detail.

Section 3.2.1 was extended as follows:

“The yield strength of sawdust particles, py was determined as the value providing the minimum RRMSE of the DEM approximation of the experimental stress–deformation relationship of the compaction process. As presented in Figure 9, the experimental relationships between sz and Dh/h0, where h0 is the initial height of the sawdust in the mould, for compaction of sawdust were very similar for the three wood materials. Therefore, a common value of py was applied for all three materials. The DEM simulation performed for py = 100 MPa fitted all the experimental relationships very well for Dh/h0 < 0.75 (RRMSE < 0.15).”

It is also recommended to revise the last paragraph of the section General remarks starting “Some problems….” Such vague expression do not belong in a scientific article.

The paragraph was corrected as follows:

“Some problems encountered during the DEM modelling of sawdust and their products may result from very limited access to verified experimental data of micro-properties of sawdust particles [51]. Therefore, more experimental data of micro-properties are necessary to fasten further development of DEM modelling of this materials.”

Reviewer 3 Report

The paper focuses on the study of wood pellets, considered an energy resource of great interest from the point of view of sustainability. Its mechanical strength is analyzed, as a relevant index of its durability, both experimentally and by numerical modeling.

The document complements the research previously published in Materials, through two papers (references numbers 42 and 53) with which it shares several authors. The one under review presents a work of great interest in its field, including aspects that can be considered novel, especially in the treatment of simulations.

Three species of wood are analyzed (oak, pine and birch), with two moisture contents (8 and 20%), and pelletized with two compaction pressures (60 and 120 MPa). The materials are adequately described, as well as the methods used in the determination of the different parameters involved both in the pellet compaction process and in the study of strength by diametrical compression test.

Title and abstract are appropriate to the research, so that the latter constitutes an illustrative synthesis of the contents. The conclusions provide useful considerations, although, as is usual in these topics, the need for large and verified sets of experimental results is highlighted. Likewise, the references are relevant and representative of the state of the art.

Therefore, in the opinion of this reviewer, the paper is suitable for publication in the journal. In any case, several doubts and minor suggestions are provided in order to improve the final manuscript.

As has been established, the experimental research comprises twelve possible combinations (three woods, 2 moisture contents, and 2 compaction pressures). However, the number of specimens tested in each category is not clearly determined. As has been established, the experimental investigation comprises twelve possible combinations (three woods, two moisture contents, and two compaction pressures). However, the number of specimens tested in each category is not clearly determined.

In 2.1.3 five replicates are mentioned in the determination of the solid-phase density. The same value appears in 2.1.5 regarding the use of scanning electron microscopy. Figures 2 and 3 describe averages curves for two replicates (two moisture contents per wood species?). It would be convenient to clarify the number of specimens studied in each phase and category, considering that a certain scattering of results may occur.

Page 3, line 122. V is expressed in m-3.kg-1. Meters must not have a negative exponent: m3.kg-1

Page 4, line 148. The variable t is defined as thickness. Perhaps it would be more appropriate to designate it as the height of the cylinder that makes up the pellet.

Page 5, line 174. The table footer is assumed to match that of tables 2 and 3.

Page 8, figure 5a. It is recommended to improve the quality of the graph, and perhaps expand the area where the compaction and unloading plots are concentrated, to facilitate its interpretation.

Page 10, equation (10). Subscript and superscript are interchanged in the velocity-dependent dissipative component (according to the text and as indicated in the EDEM Reference Guide and other references): ftd

Page 13, table 4. The restitution coefficient should be designated as e, rather than E.

Author Response

Dear Reviewer,

Thank you very much for reviewing our manuscript and for your valuable comments, which helped us to improve the manuscript. All comments were considered. Introduced changes were highlighted in the manuscript.

The paper focuses on the study of wood pellets, considered an energy resource of great interest from the point of view of sustainability. Its mechanical strength is analyzed, as a relevant index of its durability, both experimentally and by numerical modeling.

The document complements the research previously published in Materials, through two papers (references numbers 42 and 53) with which it shares several authors. The one under review presents a work of great interest in its field, including aspects that can be considered novel, especially in the treatment of simulations.

Three species of wood are analyzed (oak, pine and birch), with two moisture contents (8 and 20%), and pelletized with two compaction pressures (60 and 120 MPa). The materials are adequately described, as well as the methods used in the determination of the different parameters involved both in the pellet compaction process and in the study of strength by diametrical compression test.

Title and abstract are appropriate to the research, so that the latter constitutes an illustrative synthesis of the contents. The conclusions provide useful considerations, although, as is usual in these topics, the need for large and verified sets of experimental results is highlighted. Likewise, the references are relevant and representative of the state of the art.

Therefore, in the opinion of this reviewer, the paper is suitable for publication in the journal. In any case, several doubts and minor suggestions are provided in order to improve the final manuscript.

As has been established, the experimental research comprises twelve possible combinations (three woods, 2 moisture contents, and 2 compaction pressures). However, the number of specimens tested in each category is not clearly determined. As has been established, the experimental investigation comprises twelve possible combinations (three woods, two moisture contents, and two compaction pressures). However, the number of specimens tested in each category is not clearly determined.

In 2.1.3 five replicates are mentioned in the determination of the solid-phase density. The same value appears in 2.1.5 regarding the use of scanning electron microscopy. Figures 2 and 3 describe averages curves for two replicates (two moisture contents per wood species?). It would be convenient to clarify the number of specimens studied in each phase and category, considering that a certain scattering of results may occur.

Thank you very much for your valuable remark.

Entire experiment comprised twelve combinations (3 woods, 2 moisture contents, 2 compaction pressures). Ten specimens of each variant of pellets were prepared. All tests, except the mercury porosimetry, were performed in five replications.

The following sentences were added into the Section 2.1.2: “Twelve combinations of pellets were prepared: three woods, two moisture contents, and two compaction pressures. Ten specimens of each variant of pellets were prepared.”

Reviewer 4 Report

Dear authors,

I want to congratulate you for the comprehensive study about “Breakage Strength of Wood Sawdust Pellets: Measurements and Modelling”.

It is a manuscript about use of a by-products of woodworking industry, namely sawdust of three wood species (two hardwoods oak and birch and one softwood: pine) for the manufacture of pallets. The pellets are obtained by compaction of sawdust (binderless, as no adhesive was mentioned) and mechanical tests were performed. Additionally, a discrete element method was performed to check the numerical reproduction of palletisation.

Minor changes are required:

  1. In Introduction section, make a short link between sawdust as a lignocellulosic by-product (not waste) and its suitability for the manufacture of pelltes.
  2. Wood pellets are used since more than a decade, consequently represent no more a recent developed product (Abstract, line 11).
  3. At Page 2, line 84, add please the Latin denominations of wood species (with italics) in parentheses.
  4. Improve the quality of Figures: 5,6,11,14 and 15.

Author Response

Dear Reviewer,

Thank you very much for reviewing our manuscript and for your valuable comments, which helped us to improve the manuscript. All comments were considered. Introduced changes were highlighted in the manuscript.

I want to congratulate you for the comprehensive study about “Breakage Strength of Wood Sawdust Pellets: Measurements and Modelling”.

Thank you very much for your kind words.

It is a manuscript about use of a by-products of woodworking industry, namely sawdust of three wood species (two hardwoods oak and birch and one softwood: pine) for the manufacture of pallets. The pellets are obtained by compaction of sawdust (binderless, as no adhesive was mentioned) and mechanical tests were performed. Additionally, a discrete element method was performed to check the numerical reproduction of palletisation.

Minor changes are required:

  1. In Introduction section, make a short link between sawdust as a lignocellulosic by-product (not waste) and its suitability for the manufacture of pelltes.

The following sentence was added: “Hydrogen binding at surfaces of lignocellulosic particles of sawdust provides the main type of binding forces used in manufacturing of pellets [1].”

  1. Wood pellets are used since more than a decade, consequently represent no more a recent developed product (Abstract, line 11).

The text was corrected.

  1. At Page 2, line 84, add please the Latin denominations of wood species (with italics) in parentheses.

Latin denominations of wood species were added.

  1. Improve the quality of Figures: 5,6,11,14 and 15.

Quality of all figures was improved.